# The association between smartphone use and sleep quality, psychological distress, and loneliness among health care students and workers in Saudi Arabia

Abdullah Muhammad Alzhrani[1], Khalid Talal Aboalshamat[2], Amal Mohammmad Badawoud[3]*, Ismail Mahmoud Abdouh[4], Hatim Matooq Badri[5], Baraa Sami Quronfulah[6], Mahmoud Abdulrahman Mahmoud[7], Mona Talal Rajeh[8]

1 Department of Occupation Health, College of Public Health and Health Informatics, Umm Al-Qura University, Makkah, Saudi Arabia, 2 Preventive Dentistry Department, College of Dentistry, Umm Al-Qura University, Makkah, Saudi Arabia, 3 Department of Pharmacy Practice, College of Pharmacy, Princess Nourah bint Abdulrahman University, Riyadh, Saudi Arabia, 4 Department of Oral Basic and Clinical Sciences, College of Dentistry, Taibah University, Al Madinah Al Munawara, Saudi Arabia, 5 Department of Environmental Health, College of Public Health and Health Informatics, Umm Al-Qura University, Makkah, Saudi Arabia, 6 Department of Health Promotion and Health Education, College of Public Health and Health Informatics, Umm Al-Qura University, Makkah, Saudi Arabia, 7 College of Medicine, Imam Muhammad Ibn Saud Islamic University, Riyadh, Saudi Arabia, 8 Department of Dental Public Health, King Abdulaziz University, Faculty of Dentistry, Jeddah, Saudi Arabia

* AMBadawoud@pnu.edu.sa

**Data Availability Statement:** All relevant data are within the paper and its Supporting information files.

## Abstract

### Background

The use of smartphones among the general public and health care practitioners, in particular, is ubiquitous. The aim of this study was to investigate the relationship between smartphone addiction and sleep quality, psychological distress, and loneliness among health care students and workers in Saudi Arabia.

### Methods

This cross-sectional study used an online questionnaire to collect data on smartphone addiction, sleep quality, psychological distress, and loneliness as well as demographic information.

### Results

A total of 773 health care students and workers participated in the study, with an average age of 25.95 ± 8.35, and 59.6% female participants. The study found a positive significant association between smartphone addiction and psychological distress ($F_{(1,771)}$ = 140.8, $P <$ 0.001) and emotional loneliness ($F_{(1,771)}$ = 26.70, $P <$ 0.001). Additionally, a significant negative association between smartphone addiction and sleep quality was found ($F_{(1,771)}$ = 4.208, $P =$ 0.041). However, there was no significant relationship between smartphone addiction and social loneliness (F (1,771) = 0.544, $P <$ 0.461).

**Funding:** The author(s) received no specific funding for this work.

**Competing interests:** The authors have declared that no competing interests exist.

## Conclusion

These findings suggest that smartphone addiction has a negative impact on psychological distress, sleep quality, and emotional loneliness among health care students and workers. It is important to promote strategies to reduce smartphone dependency in order to avoid the harmful consequences of smartphone addiction.

## Introduction

The smartphone is defined as a "portable device with a touch screen interface that can be used with stylus or finger touch" [1]. Smartphones are ubiquitous among young adults and are an integral part of everyday life. The Pew Research Center reported that the ownership of smartphones among individuals ranges from 60% to 95%, and the percentage is increasing [2]. In Saudi Arabia, almost 99% of adults own smartphones [3]. The smartphone is used for communicating, including via social media, accessing information, and entertainment [4]. There are countless applications for smartphone use, including in the education and health fields. In education, the smartphone enables students to search and access information conveniently as well as use them in classrooms to view content and topics related to their studies [5, 6]. In the health care field, there are many mobile applications designed to be used by patients and health care students and professionals to improve training, education, and health outcomes [7–10]. During the COVID-19 pandemic, smartphone applications were used to trace contact with confirmed infected cases, confirm vaccination status, and implement quarantines [11]. These all emphasize the positive impacts of smartphones.

However, there are increasing concerns among researchers about the adverse health impacts smartphones could have on users. Several adverse impacts, including smartphone addiction, sleep disruption, loneliness, and distress, have been associated with smartphone use [12–15]. The use of smartphone relays on the availability of internet concoction since most of the smartphone application require internet connection. Therefore, some researchers consider smartphone addiction to be part of generalized internet addiction, when smartphones are overused to conduct a variety of activity, rather than specific internet addiction, when the internet is overused to conduct a particular activity such as social media use or gaming [16, 17]. Both, specific and generalized internet addiction have been reported to be associated with poor sleep quality, psychological distress, and loneliness [18–21].

Good sleep quality is critical to both physical and mental health and well-being. Poor sleep has been associated with adverse psychological and physical effects, such as depression, anxiety, heart disease, and psychological distress [22, 23]. One factor found to be affecting sleep quality among young adults is smartphone use [24]. Two of the proposed mechanisms of how smartphone use reduces sleep quality are the blue light from the smartphone screen interfering with melatonin levels and the electromagnetic fields affecting brain activity [25]. The association between smartphone addiction and sleep quality was investigated by Yang et al. [25] in a systematic review and meta-analysis. Their findings indicated an increased risk of poor sleep quality among smartphone addicts [25].

Psychological distress, which describes negative feelings of hopelessness and an inability to cope, has been shown to be an indicator of poor mental health and applied to a combination of undifferentiated symptoms, including anxiety, depression, behavioral problems, and functional disabilities [26]. In some studies, psychological distress was associated with smartphone addiction among university students [15, 16, 27]. Loneliness or one's perception of

discrepancy between real and desired social relationships [28] is an important determinant of health [29]. Several studies have reported a positive correlation between smartphone addiction and loneliness [30–32]. Many studies distinguish between social loneliness, as the lack of social interaction, and emotional loneliness, as the lack of intimate attachment [33].

Given the limited number of such studies in Saudi Arabia, the objective of this study was to assess whether smartphone addiction is associated with poor sleep quality and feelings of distress and loneliness among health care students and workers in Saudi Arabia. The study provides information about the prevalence of smartphone addiction and its influence on sleep quality, loneliness, and psychological distress.

## Material and methods

This cross-sectional study was part of a large observational study assessing the psychological health and lifestyle habits among health care students and workers in Saudi Arabia. Similarities in the methodology may be found among the publications of our team on this project.

The targeted participants of the study were health care students, interns, and workers. The data were taken from participants in the fields of medicine, dentistry, pharmacy, public health, and applied medical sciences. The study used a convenience sampling method to collect data from universities, governmental and private hospitals, and clinics using personal invitations and social media platforms to send the questionnaire link through professional groups in WhatsApp, Twitter, Snapchat, and other platforms. Eligibility criteria were being 1) a student, intern, or graduate in medicine, dentistry, pharmacy, public health, or applied medical sciences; 2) residing in Saudi Arabia; and 3) agreeing to the study consent form.

This study adhered to the Declaration of Helsinki and was approved by Umm Al-Qura University ethical approval committee (number: HAPO-02-K-012-2022-04-1048). The data were collected from April 6, 2022, until June 1, 2022. Prior to participation, each participant had the opportunity to ask questions related to the study, which the research team answered. Each participant provided consent prior to participation in the study. Participation was voluntary and anonymous, with no incentives. Data were stored in a private device with a password known only to members of the research team.

The questionnaire was distributed as soft copy (online format), was self-administered, and was presented in Arabic. The questionnaire comprised 33 questions in five sections measuring demographic data, smartphone addiction, psychological distress, sleep quality, and loneliness, as explained below. Section one collected demographic variables, including age, gender, qualification, specialty, region, city, and nationality and contained yes/no questions about the presence of chronic disease, walking regularly, and eating healthy foods. The second section assessed smartphone addiction using the six-item Smartphone Application-Based Addiction Scale (SABAS) [34], which comprises six statements answered on a six-point Likert scale ranging from strongly disagree (1) to strongly agree (6). The scores were totaled as a SABAS score that ranged from 6 to 36 (highest level of addiction). According to a previous study, a high score of 21 and above indicates high levels of smartphone addiction [35]. The questionnaire was translated into different languages and achieved good Cronbach's alpha scores of 0.81 to 0.88 in English [36, 37], 0.83 in Taiwanese [35], and 0.78 to 0.79 in Chinese [38].

The third section assessed psychological distress, measured by the well-known Kessler scale (K10) [39, 40]. The questionnaire is composed of 10 items using a five-point Likert scale, where 1 is none of the time, 2 is a little of the time, 3 is some of the time, 4 is most of the time, and 5 is all of the time. The answers are totaled to K10 scores that range from 5 to 50 (highest level of psychological distress) [41]. The scale has good psychometric properties, with a Cronbach's alpha of 0.88 [42]. The scale's validated Arabic version was used in this study [43].

The fourth section assessed sleep quality using the one-item Sleep Quality Scale (SQS) [44], asking, "During the past 7 days, how would you rate your sleep quality overall?" Participants could answer from 0 (terrible) to 10 (excellent). The SQS had a good relationship to the morning questionnaire-insomnia Pittsburgh Sleep Quality Index, showing validity [44].

The last section, regarding loneliness, was measured by the DeJong Gierveld loneliness scale (DGLS) [33]. This scale is divided into two subscales: emotional loneliness (EL) and social loneliness (SL). The total score of DGLS is the summation of the two subscale scores, each measured by three questions with possible answers of "yes," "more or less," and "no." The Cronbach's alpha of the scale with the subscale ranged from 0.69 to 0.76 [33].

The translation from English to Arabic was conducted for SABA, SQS, EL, and SL by a team of eight authors bilingual in Arabic and English, from medicine, dentistry, public health, and pharmacy fields, to validate the translation. In the pilot phase of the project, the translation was tested for understanding to confirm that it matched the original version. The questionnaire's pilot phase used 13 participants who fulfilled the study criteria to validate the questionnaire in terms of understanding, syntax, language, grammar, and organization. The questionnaire took 10 to 15 minutes to complete.

After data collection, data entry and coding were conducted using Microsoft Excel (Microsoft Corp., Redmond, WA). The data were then transferred into SPSS software v.27 (IBM Corp., Armonk, NY) for descriptive analysis, which included frequency, percentages, mean, and standard deviation. Inferential tests used in this study were t-test, ANOVA, and linear regression as parametric tests, and Mann–Whitney and Kruskal–Wallis as nonparametric tests. Significance was set at $P < 0.05$.

## Results

The analyzed data involved 773 participants from 33 cities across Saudi Arabia: Abha, Ahsa, Badaea, Badr, Baha, Beshah, Buljurashi, Buraidah, Dahran, Dammam, Hafr Albaten, Jazan, Jeddah, Khames Mushait, Kharj, Khobar, Madinah, Makkah, Muhayel, Najran, Qassim, Qatif, Ras Tanura, Riyadh, Shaqra, Skaka, Tabok, Taif, Traif, Umluj, Unyzah, Wajh, and Yanbu. The participants had a mean age of 25.95 ± 8.35 years. Other demographic data are shown in Table 1.

On average, the participants' smartphone addiction score was 20.83 ± 5.97 on a scale from 6 to 36 points (highest addiction level). The results showed females, students, participants from the western region, and those who did not walk regularly or typically eat healthy food had smartphone addiction scores greater than 21 on average, which is the cut off score for the high smartphone addiction level [35]. As shown in Table 1, t-test, ANOVA, Mann–Whitney, and Kruskal–Wallis were used to examine the association between smartphone addiction and participants' demographic variables. The smartphone addiction score for females (21.18 ± 5.96) was significantly higher than males (20.32 ± 5.99, $P = 0.049$), Saudis (20.94 ± 5.95) was significantly higher than non-Saudis (17.58 ± 5.90, $P = 0.012$), those who do not walk regularly (21.51 ± 5.84) was significantly higher than their counterparts (19.67 ± 6.05, $P < 0.001$), and those who do not typically eat healthy foods (21.58 ± 5.68) was significantly higher than those who do (19.78 ± 6.24, $P < 0.001$; Table 1).

Using simple linear regression, there was a significant relationship ($F_{(1,771)} = 140.8$, $P < 0.001$) between smartphone addiction and psychological distress, with R-squared of 0.153 (Fig 1). There was also a significant negative relationship between smartphone addiction and sleep quality ($F_{(1,771)} = 4.208$, $P = 0.041$), with R-squared of 0.004 (Fig 2). There was also a direct significant relationship between smartphone addiction and emotional loneliness (F

**Table 1. Participant demographic data and associations with smartphone addiction.**

| Demographics | | N (%) | Smartphone addiction mean (SD) | P-value |
|---|---|---|---|---|
| **Gender** | Male | 312 (40.36) | 20.32 (5.99) | **0.049**[a] |
| | Female | 461 (59.64) | 21.18 (5.96) | |
| **Specialty** | Dentistry | 181 (23.42) | 20.8 (5.67) | 0.151[b] |
| | Public health | 268 (34.67) | 20.93 (5.67) | |
| | Pharmacy | 105 (13.58) | 20.03 (6.99) | |
| | Medicine | 161 (20.83) | 20.62 (6.19) | |
| | Applied medical science | 58 (7.50) | 22.51 (5.59) | |
| **Qualification** | Student | 435 (56.27) | 21.11 (5.82) | 0.066[b] |
| | Intern/resident | 189 (24.45) | 20.98 (5.96) | |
| | Specialist or consultant | 149 (19.28) | 19.81 (6.39) | |
| **Region** | Western | 478 (61.84) | 21.21 (5.98) | 0.285[c] |
| | Central | 193 (24.97) | 20.16 (6.09) | |
| | Southern | 71 (9.18) | 20.62 (5.78) | |
| | Eastern | 21 (2.72) | 19.14 (6.12) | |
| | Northern | 10 (1.29) | 20.64 (3.91) | |
| **Nationality** | Saudi | 747 (96.64) | 20.94 (5.95) | **0.012**[d] |
| | Non-Saudi | 26 (3.36) | 17.58 (5.9) | |
| **Do you have a chronic disease?** | Yes | 106 (13.71) | 20.47 (6.25) | 0.519[a] |
| | No | 667 (86.29) | 20.89 (5.94) | |
| **Do you walk regularly?** | Yes | 285 (36.87) | 19.67 (6.05) | **<0.001**[a] |
| | No | 488 (63.13) | 21.51 (5.84) | |
| **Do you usually eat healthy food?** | Yes | 322 (41.66) | 19.78 (6.24) | **<0.001**[a] |
| | No | 451 (58.34) | 21.58 (5.68) | |

[a] t-test,

[b] ANOVA,

[c] Kruskal Wallis Test,

[d] Mann-Whitney Test

$_{(1,771)}$ = 26.70, $P < 0.001$), with R-squared of 0.032 (Fig 3). However, there was no significant relationship between smartphone addiction and social loneliness (F $_{(1,771)}$ = 0.544, $P < 0.461$).

## Discussion

The objective of this cross-sectional study was to investigate the relationship between smartphone addiction and psychological distress, sleep quality, and loneliness among health care students and workers. This study showed that the smartphone addiction mean score was 20.83 ± 5.97, which is very close to the cutoff point (21) for high addiction level [35]. The simple linear regression conducted in this study suggests that smartphone addiction is positively associated with psychological distress and emotional loneliness, while it is negatively associated with sleep quality. However, smartphone addiction was not associated with social loneliness. Thus, the results indicate that some of the participants may have smartphone addiction, especially those who are female, students, and Saudi, as well as those who do not walk or exercise regularly and do not eat healthy foods. Additionally, the findings confirmed the negative impact of smartphone addiction on mental and physical health among our sample in Saudi Arabia.

In this study, the smartphone addiction score was significantly higher among females than males, which is consistent with previous studies. The study by Tateno et al. [45]. among college

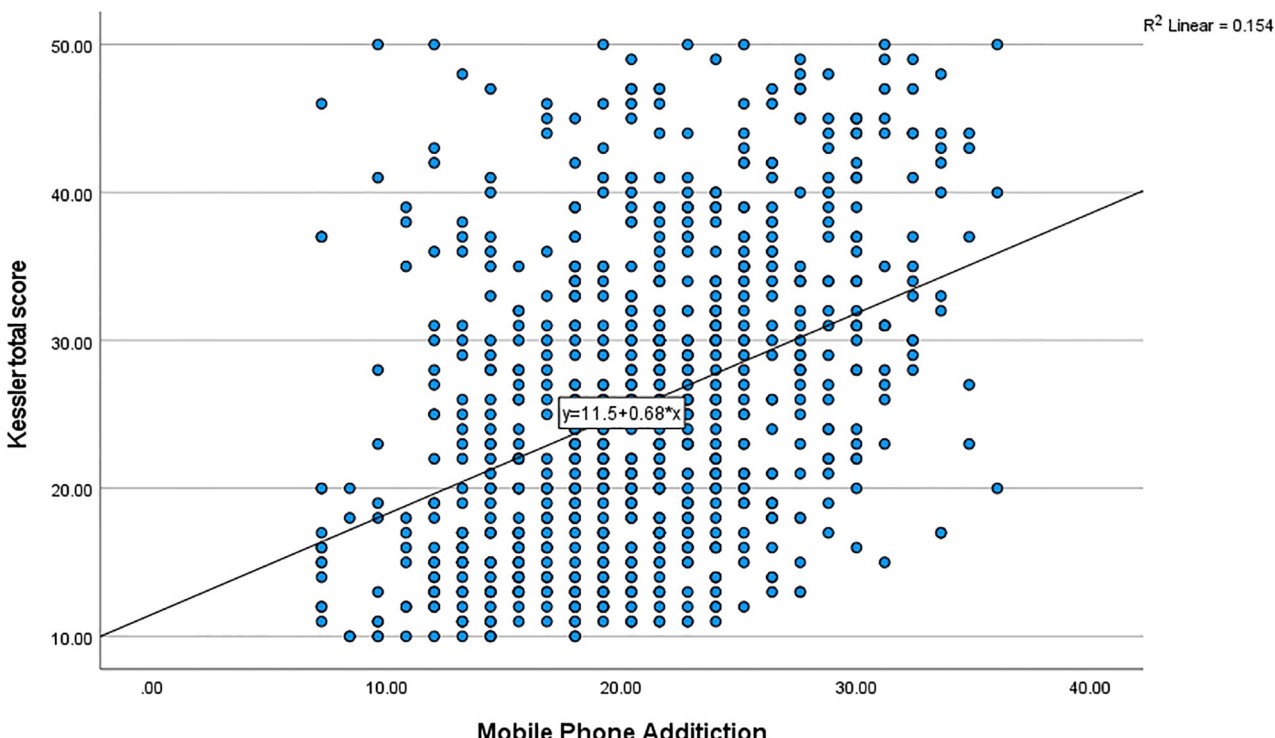

**Fig 1. The relationship between mobile phone addiction and psychological distress among health professionals.**

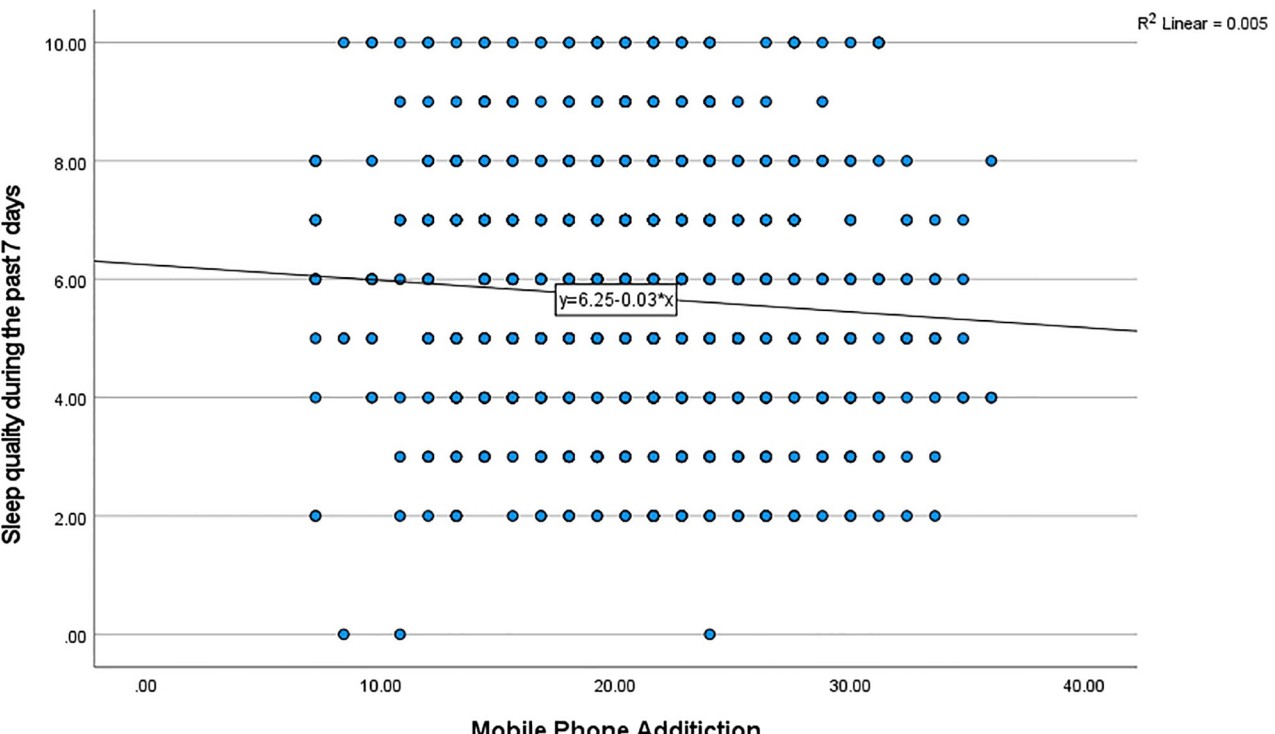

**Fig 2. The relationship between mobile phone addiction and sleep quality.**

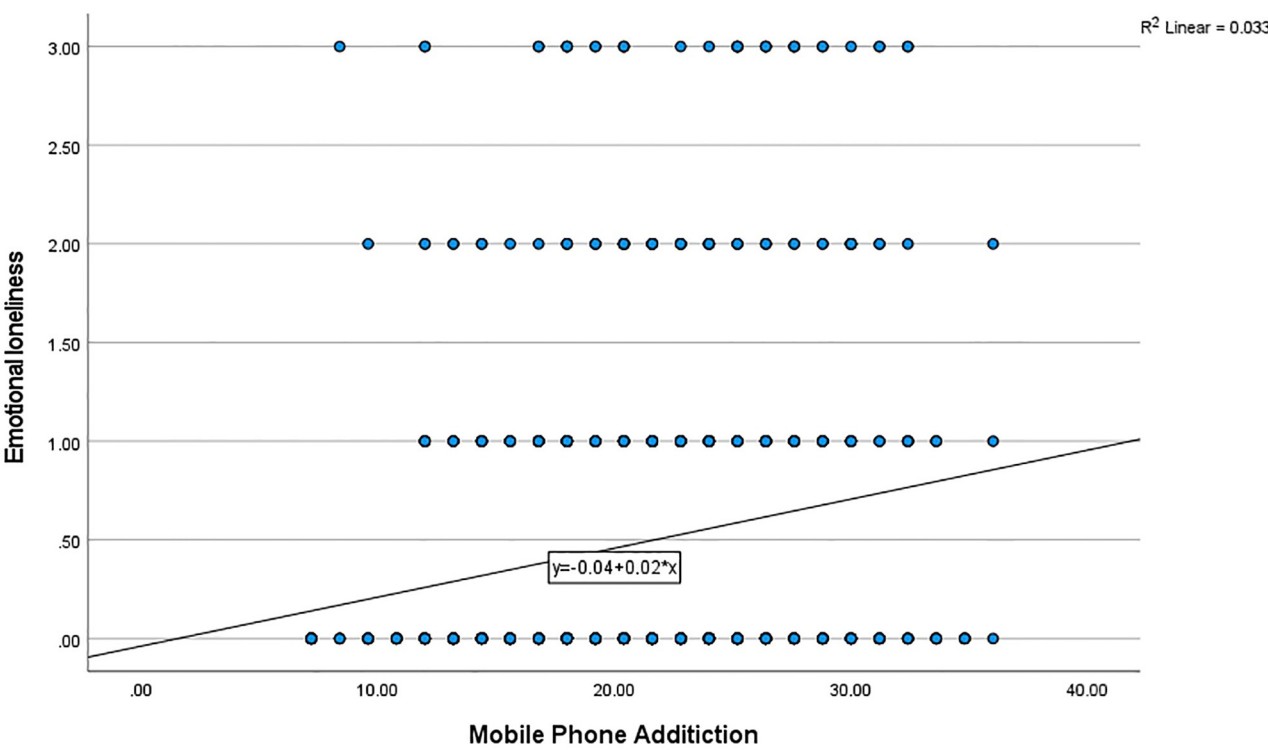

**Fig 3. The relationship between mobile phone addiction and emotional loneliness.**

students reported that females had higher smartphone addiction scores than male students. Hegazy et al. studied the use of smartphones among medical students at a Saudi university and found that females spent longer durations using their smartphones than males [46]. Moreover, females were found to be more dependent on their smartphones than males [47–49]. These findings suggest that females may be at greater risk of experiencing negative impacts from smartphone addiction on their physical and mental health.

The findings of this study indicate that higher smartphone addiction leads to poorer sleep quality, which may impact the overall health of the user. Several studies reported similar findings. Kumar et al. [50] reported that smartphone addiction was associated with poor sleep quality among medical students. In another study, smartphone addiction was associated with poor sleep quality and lower academic performance [48]. This suggests that smartphone addiction could directly negatively influence academic and work performance by distracting health care students and workers from their studies or work. Smartphone addiction may also negatively impact sleep quality, which may lead to poorer academic and work performance. This indicates the importance of intervention to reduce smartphone addiction among health care students and workers to improve overall sleep quality and avoid negative impacts of smartphone addiction.

Psychological distress was positively associated with smartphone addiction among our participants. This indicate that the more a participant is addicted to smartphone use, the more likely they are to suffer from psychological distress. This finding is consistent with other studies on smartphone addiction and psychological distress [15, 16, 27]. Chen et al. [27] reported that problematic smartphone use is a risk factor that may lead to psychological distress. A meta-analysis reported that smartphone addiction was associated with greater social distress

levels among medical students [14]. However, another study suggested that persons experiencing psychological distress use their smartphone more often to decamp from negative feelings due to distress, indicating that distress may be an antecedent of smartphone addiction [13]. The current study and literature echo the association between smartphone addiction and psychological distress, which may lead to several negative implications that include, but are not limited to, attrition from study or work, impaired performance, problematic health behaviors, anxiety, panic disorder, and depression [51].

Although the study did not find a significant association between smartphone addiction and social loneliness, there was a positive association between smartphone addiction and emotional loneliness. This is a discrepancy with other studies in the literature where smartphone addiction was associated with both social and emotional loneliness [52] or loneliness in general [12, 30, 32]. Mahapatra [12] reported that individuals who feel lonely tend to use their smartphone more, which may consequently lead to personal conflict and poor performance. However, other studies suggested that the relationship between smartphone addiction and feeling lonely is a loop, where individuals get disconnected from people around them due to their increased use of smartphones, and then they feel lonely. Due to feeling lonely, they spend even more time on their smartphone, and so on [52]. In this study, a possible explanation of our finding is that our participants used their smartphones to stay connected with other people using social media, thus satisfying their need to be socially connected. However, this connection with people via social media may not fulfill emotional needs due to lack of close contact with friends, which leads to the feeling of emotional loneliness. Loneliness has been linked to several physical and mental consequences, including cardiovascular disease, obesity, anxiety, and depression [53].

As indicated by the findings of this study and others, it is recommended to promote strategies to reduce smartphone dependency among health care students and workers. A meta-analysis by Malinauskas and Malinauskiene [54] concluded that cognitive behavioral therapy intervention is an effective method to reduce smartphone addiction. Another meta-analysis reported that increasing physical activity through exercise intervention could have positive effects on the management of smartphone addiction, especially when the smartphone user exercises on regular and continuous bases [55]. This is supported by this study, as we found that those who walk regularly had significantly lower addiction scores than those who do not walk or exercise. Educating smartphone addicts regarding coping strategies may also reduce addictive smartphone behavior [52]. It is also critical to manage withdrawal symptoms, such as negative feelings, impulsiveness, and anxiety, using psychological evaluations when necessary [56].

There are some limitations in this study's design and findings. The cross-sectional design may not reflect the real-life situation, but still provides some useful information about smartphone addiction and its relationship with sleep quality, psychological distress, and loneliness. Longitudinal studies or studies with multiple follow-ups are needed to obtain more rigorous information. The second limitation is related to the generalizability of the findings to the wider population. This study focused on young health care students and workers using convenience sampling, which hinders causal inference and generalization of findings. Future studies should recruit members of the general public using random sampling to avoid this limitation. Additionally, we did not adjust for confounding factors such as age, gender, and other factors that may mediate or confound the relationship between smartphone addiction and physiological distress, sleep quality, and loneliness. As we are in COVID-19 aftermath, it is also possible that our findings are affected by the pandemic. Many studies reported association between COVID-19 and sleep and psychological problems [57, 58].

## Conclusion

Participants who were female, Saudi, health care students, and do not exercise or eat healthy foods had higher levels of smartphone addiction than others. The findings suggest that smartphone addiction is negatively associated with sleep quality and positively associated with psychological distress and emotional loneliness. Smartphone addiction should be monitored and reduced through education, exercise, and cognitive behavioral therapy interventions to lessen its effect and implications. It is also recommended to address temporality between smartphone addiction, psychological distress, sleep quality, and loneliness in future research.

## Supporting information

**S1 Dataset. K54-1 dataset.**
(XLSX)

**S1 File. K54-1 questionnaire.**
(PDF)

## Author Contributions

**Conceptualization:** Abdullah Muhammad Alzhrani, Khalid Talal Aboalshamat, Amal Mohammmad Badawoud, Ismail Mahmoud Abdouh, Hatim Matooq Badri, Baraa Sami Quronfulah, Mahmoud Abdulrahman Mahmoud, Mona Talal Rajeh.

**Data curation:** Abdullah Muhammad Alzhrani, Khalid Talal Aboalshamat, Amal Mohammmad Badawoud, Ismail Mahmoud Abdouh, Hatim Matooq Badri, Baraa Sami Quronfulah, Mahmoud Abdulrahman Mahmoud, Mona Talal Rajeh.

**Formal analysis:** Khalid Talal Aboalshamat.

**Project administration:** Khalid Talal Aboalshamat.

**Writing – original draft:** Abdullah Muhammad Alzhrani.

**Writing – review & editing:** Khalid Talal Aboalshamat, Amal Mohammmad Badawoud, Ismail Mahmoud Abdouh, Hatim Matooq Badri, Baraa Sami Quronfulah, Mahmoud Abdulrahman Mahmoud, Mona Talal Rajeh.

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
