## [Decision Letter · Decision Letter 0]

29 Nov 2022

PONE-D-22-22716The association between smartphone use and sleep quality, psychological distress, and loneliness among health care students and workers in Saudi ArabiaPLOS ONE

Dear colleagues, 

Thank you for submitting your manuscript to PLOS ONE. After careful consideration, we feel that it has merit but does not fully meet PLOS ONE’s publication criteria as it currently stands. Therefore, we invite you to submit a revised version of the manuscript that addresses the points raised during the review process.

We look forward to receiving your revised manuscript.

Kind regards,

Yaser Mohammed Al-Worafi

Academic Editor

PLOS ONE

Journal Requirements:

2. Please ensure that you include a title page within your main document. You should list all authors and all affiliations as per our author instructions and clearly indicate the corresponding author.

Reviewers' comments:

Reviewer's Responses to Questions

**Comments to the Author**

1. Is the manuscript technically sound, and do the data support the conclusions?

Reviewer #1: Yes

Reviewer #2: Partly

2. Has the statistical analysis been performed appropriately and rigorously? 

Reviewer #1: Yes

Reviewer #2: I Don't Know

3. Have the authors made all data underlying the findings in their manuscript fully available?

Reviewer #1: Yes

Reviewer #2: No

4. Is the manuscript presented in an intelligible fashion and written in standard English?

Reviewer #1: Yes

Reviewer #2: Yes

5. Review Comments to the Author

Reviewer #1: The study entitled "The association between smartphone use and sleep quality, psychological distress, and loneliness among health care students and workers in Saudi Arabia" has good sample size (n=773), a meaningful sample (i.e., healthcare students and workers), and appropriate statistical analyses. However, it should be improved with the following considerations.

1. Please elaborate more on the literature. The authors are encouraged in describing the generalized internet addiction and specific internet addiction in the Introduction. Moreover, the authors can indicate that some scholars consider smartphone addiction as generalized internet addiction because smartphone provides a variety of activities instead of specific activities (e.g., social media use and gaming).

2. The authors then should use some references to indicate that both generalized internet addiction and specific internet addiction have been found to be associated with sleep problems or psychological distress among students. This strengthen the rationale of studying the association between smartphone use and sleep quality/psychological distress.

Ranjan LK, Gupta PR, Srivastava M, Gujar NM. Problematic internet use and its association with anxiety among undergraduate students. Asian J Soc Health Behav 2021;4:137-41

Oluwole LO, Obadeji A, Dada MU. Surfing over masked distress: Internet addiction and psychological well-being among a population of medical students. Asian J Soc Health Behav 2021;4:56-61

Kwok C, Leung PY, Poon KY, Fung XC. The effects of internet gaming and social media use on physical activity, sleep, quality of life, and academic performance among university students in Hong Kong: A preliminary study. Asian J Soc Health Behav 2021;4:36-44

Patel VK, Chaudhary P, Kumar P, Vasavada DA, Tiwari DS. A study of correlates of social networking site addiction among the undergraduate health professionals. Asian J Soc Health Behav 2021;4:30-5

3. It would be better to mention the association between sleep and psychological distress.

Pengpid S, Peltzer K. Anxiety-induced sleep disturbances among in-school adolescents in the United Arab Emirates: Prevalence and associated factors. Asian J Soc Health Behav 2022;5:18-23

4. In Introduction, the authors have mentioned the impacts of COVID-19. They may want to elaborate this using some recent publications in the Sleep Epidemiology.

5. The SABAS has been recently validated in Malaysia student population. The authors may want to mention this to strengthen the rationale of using SABAS in the present study.

Tung, S. E. H., Gan, W. Y., Chen, J.-S., Kamolthip, R., Pramukti, I., Nadhiroh, S. R., Chang, Y.-L., Lin, C.-C., Pakpour, A. H., Lin, C.-Y., Griffiths, M. D. (2022). Internet-related instruments (Bergen Social Media Addiction Scale, Smartphone Application-Based Addiction Scale, Internet Gaming Disorder Scale-Short Form, and Nomophobia Questionnaire) and their associations with distress among Malaysian university student. Healthcare, 10, 1448.

6. The statement "Moreover, females were found to be more dependent on their smartphones than males" can also be supported by a recent study.

Xu P, Chen JS, Chang YL, et al. Gender Differences in the Associations Between Physical Activity, Smartphone Use, and Weight Stigma. Front Public Health. 2022;10:862829. Published 2022 Mar 29. doi:10.3389/fpubh.2022.862829

Reviewer #2: The first comment is you have not added line numbers for your document, it makes the comments to be addressed accurately.

The figure 1 is not clear at all, also could not find figures 2 and 3. Table 1 does not present t-test or ANOVA as you claimed at the end of your methodology.

You have presented interesting results, but you have no recommendation or suggestion. I think you should add a paragraph at least.

I think you need to recheck carefully.

6. PLOS authors have the option to publish the peer review history of their article (what does this mean?). If published, this will include your full peer review and any attached files.

Reviewer #1: No

Reviewer #2: **Yes: **Nazdar Qudrat Abas

---

## [Author Response · Author response to Decision Letter 0]

2 Jan 2023

Respond to reviewer

We would like to thank the reviewers for their insightful comments, which we believe have overall improved the manuscript. In the below table, we have addressed each comment individually. To be organized, we used R1 and R2 to indicate the reviewer and a number for the comment (e.g. R1-1, R1-2 and so on)

Reviewer#1

R1-1: The study entitled "The association between smartphone use and sleep quality, psychological distress, and loneliness among health care students and workers in Saudi Arabia" has good sample size (n=773), a meaningful sample (i.e., healthcare students and workers), and appropriate statistical analyses. However, it should be improved with the following considerations.

Response: Thank you. Each comment was considered and has been addressed

R1-2: 1. Please elaborate more on the literature. The authors are encouraged in describing the generalized internet addiction and specific internet addiction in the Introduction. Moreover, the authors can indicate that some scholars consider smartphone addiction as generalized internet addiction because smartphone provides a variety of activities instead of specific activities (e.g., social media use and gaming).

Response: To address this comment additional information about generalized and specific smartphone/internet addition was added to the introduction along with appropriate references as follows:

Change:

…, including smartphone addiction, sleep disruption, loneliness, and distress, have been associated with smartphone use.[12–15] The use of smartphone relays on the availability of internet concoction since most of the smartphone application require internet connection. Therefore, some researchers consider smartphone addiction to be part of generalized internet addiction, when smartphones are overused to conduct a variety of activity, rather than specific internet addiction, when the internet is overused to conduct a particular activity such as social media use or gaming. [16,17] Both, specific and generalized internet addiction have been reported to be associated with poor sleep quality, psychological distress, and loneliness. [18–21]

R1-3: The authors then should use some references to indicate that both generalized internet addiction and specific internet addiction have been found to be associated with sleep problems or psychological distress among students. This strengthen the rationale of studying the association between smartphone use and sleep quality/psychological distress.

Ranjan LK, Gupta PR, Srivastava M, Gujar NM.Problematic internet use and its association with anxiety among undergraduate students. Asian J Soc Health Behav 2021;4:137-41

Oluwole LO, Obadeji A, Dada MU.Surfing over masked distress: Internet addiction and psychological well-being among a population of medical students. Asian J Soc Health Behav 2021;4:56-61 

Kwok C, Leung PY, Poon KY, Fung XC. The effects of internet gaming and social media use on physical activity, sleep, quality of life, and academic performance among university students in Hong Kong: A preliminary study. Asian J Soc Health Behav 2021;4:36-44 

Patel VK, Chaudhary P, Kumar P, Vasavada DA, Tiwari DS. A study of correlates of social networking site addiction among the undergraduate health professionals. Asian J Soc Health Behav 2021;4:30-5

Response: To address this comment additional information about generalized and specific smartphone/internet addition was added to the introduction along with the suggested relevant references as follows:

Changes: 

“Smartphone addiction has been reported to be associated with poor sleep quality, psychological distress, and loneliness. [1–4]

R1-4: It would be better to mention the association between sleep and psychological distress.

Pengpid S, Peltzer K. Anxiety-induced sleep disturbances among in-school adolescents in the United Arab Emirates: Prevalence and associated factors. Asian J Soc Health Behav 2022;5:18-23

Response: We agree to this comment and text was edited

Changes: 

 “Poor sleep has been associated with adverse psychological and physical effects, such as depression, anxiety, heart disease, and psychological distress.[5,6]

R1-5: In Introduction, the authors have mentioned the impacts of COVID-19. They may want to elaborate this using some recent publications in the Sleep Epidemiology.

Response: We appreciate this comment. We have not mentioned the impacts of COVID-19 in this manuscript. Instead, we have gave an example of how smartphone-based applications can be useful in tracking COVID-19 cases and vaccination. 

We believe that there is no need to add more about COVID-19 impacts on sleep in the introduction because it is not the focus of this research. However, we think it is important to mention that COVID-19 may have an impact on our findings in the limitation section as follows. 

Changes: 

“As we are in COVID-19 aftermath, it is also possible that our findings are affected by the pandemic. Many studies reported association between COVID-19 and sleep and psychological problems [7,8].”

R1-6: The SABAS has been recently validated in Malaysia student population. The authors may want to mention this to strengthen the rationale of using SABAS in the present study.

Tung, S. E. H., Gan, W. Y., Chen, J.-S., Kamolthip, R., Pramukti, I., Nadhiroh, S. R., Chang, Y.-L., Lin, C.-C., Pakpour, A. H., Lin, C.-Y., Griffiths, M. D. (2022). Internet-related instruments (Bergen Social Media Addiction Scale, Smartphone Application-Based Addiction Scale, Internet Gaming Disorder Scale-Short Form, and Nomophobia Questionnaire) and their associations with distress among Malaysian university student. Healthcare, 10, 1448

Response: This study is relevant and has been added to the methods

Changes: 

“achieved good Cronbach’s alpha scores of 0.81-0.88 in English,[9] [10]”

R1-7: The statement "Moreover, females were found to be more dependent on their smartphones than males" can also be supported by a recent study.

Xu P, Chen JS, Chang YL, et al. Gender Differences in the Associations Between Physical Activity, Smartphone Use, and Weight Stigma. Front Public Health. 2022;10:862829. Published 2022 Mar 29. doi:10.3389/fpubh.2022.862829

Response: The reference is relevant and has been cited 

Changes: 

Reference was added to text and reference list

Reviewer#2

R2-1: Reviewer #2: The first comment is you have not added line numbers for your document, it makes the comments to be addressed accurately.

I think you need to recheck carefully.

Response: Due to journal submission process, line numbers was not added initially as it was not required. We apologize for any inconvenience. 

Changes: 

We added line number to the revised manuscript

R2-2: The figure 1 is not clear at all, also could not find figures 2 and 3. 

Response: We now have submitted figures with higher quality. 

Figures 2 and 3 was not submitted due to error at our end. We apologize for that. We will make sure the complete files are submitted

R2-3: Table 1 does not present t-test or ANOVA as you claimed at the end of your methodology.

Response: The P-values in Table one are actually the findings of t-test and ANOVA or Mann-Whiteny and Kruskal-Wallis tests when indicated by Asterisk (*). 

Changes:

To make this clearer, we have added footnote to indicate which statistical test was used to obtain p-values (t-test, ANOVA, Whiteny and Kruskal-Wallis tests)

R2-4: You have presented interesting results, but you have no recommendation or suggestion. I think you should add a paragraph at least.

Response: 

We appreciate the reviewer comment, however, we would like to highlight that we have provided recommendation in the 6th paragraph in the discussion section, which mention the need to promote different strategies such as education, coping, psychological assessment to manage smartphone addiction. The following is the 6th paragraph in the discussion section

“As indicated by the findings of this study and others, it is important to promote strategies to reduce smartphone dependency among health care students and workers. A meta-analysis by Malinauskas and Malinauskiene [52] concluded that cognitive behavioral therapy intervention is an effective method to reduce smartphone addiction. Another meta-analysis reported that increasing physical activity through exercise intervention could have positive effects on the management of smartphone addiction, especially when the smartphone user exercises on regular and continuous bases.[53] This is supported by this study, as we found that those who walk regularly had significantly lower addiction scores than those who do not walk or exercise. Educating smartphone addicts regarding coping strategies may also reduce addictive smartphone behavior.[50] It is also critical to manage withdrawal symptoms, such as negative feelings, impulsiveness, and anxiety, using psychological evaluations when necessary.[54]

However, to further clarify this, we have amended the first sentence to be:

Changes: 

“it is recommended to promote strategies…”

---

## [Editor Report · Decision Letter 1]

6 Jan 2023

The association between smartphone use and sleep quality, psychological distress, and loneliness among health care students and workers in Saudi Arabia

PONE-D-22-22716R1

Dear Dr. Amal, 

We’re pleased to inform you that your manuscript has been judged scientifically suitable for publication and will be formally accepted for publication once it meets all outstanding technical requirements.

Kind regards,

Yaser Mohammed Al-Worafi

Academic Editor

PLOS ONE
---

## [Editor Report · Acceptance letter]

18 Jan 2023

PONE-D-22-22716R1 

The association between smartphone use and sleep quality, psychological distress, and loneliness among health care students and workers in Saudi Arabia 

Dear Dr. Badawoud:

I'm pleased to inform you that your manuscript has been deemed suitable for publication in PLOS ONE. Congratulations! Your manuscript is now with our production department. 

Kind regards, 

on behalf of

Professor Yaser Mohammed Al-Worafi 

Academic Editor

PLOS ONE